# Hypolipemiant Actions and Possible Cardioprotective Effects of Valine and Leucine: An Experimental Study

**DOI:** 10.3390/medicina57030239

**Published:** 2021-03-05

**Authors:** Elena Cojocaru, Maria Magdalena Leon-Constantin, Carmen Ungureanu, Mioara Florentina Trandafirescu, Alexandra Maștaleru, Laura Mihaela Trandafir, Florin Dumitru Petrariu, Oana Viola Bădulescu, Nina Filip

**Affiliations:** 1Department of Morphofunctional Sciences I, “Grigore T. Popa” University of Medicine and Pharmacy, 700115 Iaşi, Romania; ellacojocaru@yahoo.com (E.C.); carmens.ungureanu@yahoo.com (C.U.); mioaratrandafirescu@yahoo.co.uk (M.F.T.); 2Department of Medical Specialties I, “Grigore T. Popa” University of Medicine and Pharmacy, 700115 Iaşi, Romania; 3Department of Mother and Child Medicine, “Grigore T. Popa” University of Medicine and Pharmacy, 700115 Iaşi, Romania; trandafirlaura@yahoo.com; 4Department of Preventive Medicine and Interdisciplinarity, “Grigore T. Popa” University of Medicine and Pharmacy, 700115 Iaşi, Romania; florin.petrariu@umfiasi.ro; 5Department of Morphofunctional Sciences II, “Grigore T. Popa” University of Medicine and Pharmacy, 700115 Iaşi, Romania; violabadulescu@yahoo.com (O.V.B.); zamosteanu_nina@yahoo.com (N.F.)

**Keywords:** atherosclerosis, valine, leucine, triglycerides, rat

## Abstract

*Background and Objectives:* Considering atherosclerosis as one of the more challenging threats to healthcare worldwide, any novel therapy that counteracts the risks for developing it, provides new opportunities for the management of this process. *Material and methods:* We performed an experimental research in which we induced a hypercholesterolemia via a cholesterol-rich diet. Our aim was to demonstrate the antiatherogenic potential of two essential amino acids (valine and leucine). The experimental study was carried out over a period of 60 days. Male Wistar rats weighing between 250–280 g were used and divided into 4 groups, each group including 8 animals. Group I—control was fed with a standard diet. Group II received cholesterol, group III cholesterol and valine and group IV cholesterol and leucine. Blood samples were collected from the retro-orbital plexus, under anesthesia with 75 mg/kg of intraperitoneal ketamine, in three different moments (R0—1st day, R1—the 30th day, R2—the 60th day) in order to measure the levels of triglycerides. *Results:* In R0, there were no significant differences between the average levels of triglycerides across all the groups (*p* < 0.05). Compared to the group I, in R1 and R2, the average levels of triglycerides were significantly higher in all groups (*p* < 0.001). Also, in R1 and R2, the average triglycerides in group II receiving cholesterol (C) were significantly higher than those in group III receiving valine (C + V) as well as in group IV receiving leucine (C + L) (*p* < 0.001; *p* < 0.05). In R2, the average triglycerides in group III were significantly lower than in group IV (*p* < 0.001). *Conclusions:* Our data provides evidence that valine and leucine have a direct impact on the lipid metabolism parameters by lowering the level of triglycerides. The comparison of the two essential amino acids indicates that valine acts more promptly and rapidly than leucine.

## 1. Introduction

Atherosclerotic cardiovascular disease represents nowadays a leading cause of death worldwide [1,2]. Surveys data of coronary patients prove that the implementation of guidelines regarding cardiovascular diseases prevention in clinical practice needs improvement [2,3]. Scientific studies related to atherosclerosis are being constantly published, in time with an increase in morbidity and mortality due to ischemic heart disease. The medical community is justifiably focused on cardiovascular diseases prevention, following the control of risk factors involved in the etiology of these disease (total cholesterol, LDL (low-density lipoprotein)-cholesterol, glycemia, uric acid, smoking, hypertension, hyperhomocysteinemia, etc.) [4,5,6].

Recent epidemiological data support the idea that hypertriglyceridemia is a prevalent risk factor for cardiovascular diseases, and plays an important role in the pathogenesis of atherosclerosis. Consequently, current perspectives of how dyslipidemia is treated should be renewed to also target the normalization of triglyceride levels [7]. Upon correction of serum LDL, both with medication as well as via a hygienic-dietetic regimen, there is still a residual risk of atherosclerosis mainly due to the lipoproteins which are known to transport triglycerides [8]. The mechanism by which lowering the level of plasma triglycerides leads to a decrease of the total cardiovascular risk appears to be the apoB (apolipoprotein B) reduction [9].

The fact that triglycerides stimulate the production of inflammatory cytokines, of fibrinogen, and coagulation factors, thus affecting fibrinolysis, is another confirmation of its role in the atherosclerotic process [10]. The involvement of triglycerides in atherosclerosis and in increasing the total cardiovascular risk is therefore evident, but additional research is necessary in order to establish if elevated levels of serum triglycerides cause ischemic heart disease per se or by association with other known cardiovascular risk factors such as diabetes mellitus or obesity [11,12].

In previous studies, we investigated the anti-atherogenic potential of valine and leucine, two non-polar amino acids, in the context of hypercholesterolemia induced by a cholesterol-rich diet, and we followed their effects on lipid metabolism (total cholesterol, HDL (high-density lipoprotein)-cholesterol, LDL-cholesterol) and oxidative stress parameters [13,14,15,16].

In this paper, we focus on our results regarding the study of triglycerides levels, agreeing that hypertriglyceridemia could become an important therapeutic target in the management of atherosclerosis.

## 2. Materials and Methods

The experimental study carried out over a period of 60 days. Male Wistar rats weighing 250–280 g were obtained from the animal farm of the “Grigore T. Popa” University of Medicine and Pharmacy, Iași. All animal protocols were carried out in accordance with the instructions of the Guide regarding animal care and scientific use, in strict accordance to international ethical regulations [17,18].

The rats were divided into four groups as follows:Control group I (*n* = 8): Fed with a regular diet composed of agricultural byproducts.Group II—C (*n* = 8): Received regular diet supplemented with 0.4 g/kg/day cholesterol.Group III—C + V (*n* = 8): Received regular diet supplemented with 0.4 g/kg/day cholesterol and 62.5 mg/kg/day valine powder for animal nutrition.Group IV—C + L (*n* = 8): Received regular diet supplemented with 0.4 g/kg/day cholesterol and 69.985 mg/kg/day leucine powder for animal nutrition.

Blood samples were collected from the retro-orbital plexus, under anaesthesia of animals with 75 mg/kg of intraperitoneal ketamine, in three moments of the experiment as follows: R0—1st day, R1—30th day R2—60th day. The measurement of triglycerides levels was made using Diagnosticum Zrt kit bought from Budapest, Hungary, as in our previous studies [19].

The study was conducted in accordance with the 2010/63/EU directive and followed the recommendations of the National Institutes of Health (NIH) Guide for the Care and the Use of Laboratory Animals. Prior to the beginning of the study, the protocol received ethical approval from the ethics committee of the University of Medicine and Pharmacy “Grigore T. Popa”, Iaşi, Ethics Committee approval number 15186/2008.

The data were centralized in Excel and SPSS databases and processed with their respective suitable statistical. Statistical confidence intervals with a 95% significance level were used. In order to evaluate the statistically significance between our groups we used ANOVA test (including repeated measures ANOVA). The threshold for statistical significance is the maximum level of probability which affords an error. A significance level of 0.05% points to sufficient precision in practice, and 95% probability indicates reliability.

## 3. Results

For each group, triglycerides were measured in R0, R1 and R2. Table 1 and Figure 1 show the levels of triglycerides throughout the experiment, as well as the variance of the measured values series. In the control group, the variance was low, ranging from 3.27 to 3.77 CV (coefficient of variation) %, with R2 results being the most homogenous (3.27 CV%). In groups II, III and IV, the coefficients of variation ranged from 2.45 to 14.82 CV%. In these cases, the most homogenous results were registered in R2 in group II—C (8.04 CV%) and group IV—C + L (4.36 CV%), and in R1 in group III—C + V (2.45 CV%).

The highest individual levels of triglycerides were found in group II corresponding to the rats who received cholesterol only (Figure 1).

A one-way repeated measure analysis of variance (ANOVA) was conducted to evaluate the null hypothesis that there is no change in the triglycerides values in different subgroups measured in R0, R1 and R2 (*N* = 32) (Table 2). The results of the ANOVA indicated a significant time effect, Wilks’ Lambda = 0.015, F (2, 27) = 879.33, *p* < 0.01, η^2^ = 0.98. Thus, there is significant evidence to reject the null hypothesis.

Follow-up comparisons indicate that each pairwise difference was significant, *p* < 0.01. There was a significant increase of values over time (Table 3).

In R0, there were no significant differences between the average levels of triglycerides in groups II, III and IV and neither between these and the control group (*p* < 0.05). Compared to the control group, the average levels of triglycerides were significantly higher in all groups in R1 and R2 (*p* < 0.001).

Also, in R1 and R2 measurements, the average triglycerides in group II receiving cholesterol only (C) were significantly higher than those in group III receiving valine (C + V) as well as in group IV receiving leucine (C + L) (*p* < 0.001; *p* < 0.05).

At the end of the experiment (R2), the average triglycerides in group III were significantly lower than in the case of rats who received leucine (*p* < 0.001) (Table 4, Figure 2).

As described before, in previous studies, we evaluated the anti-atherogenic potential of valine and leucine in the context of hypercholesterolemia induced by a cholesterol-rich diet, following their effects on lipid metabolism (total cholesterol, HDL-cholesterol, LDL-cholesterol) and oxidative stress parameters [13,14,15,16]. The mean values can be seen in Table 5. Our results showed that valine and leucine increased the serum levels of HDL-cholesterol. More specifically, after one month and at the end of the experiment, the HDL-cholesterol values in animals who received only cholesterol (C) were significantly lower compared with group III who received cholesterol and valine (C + V) or group IV that received cholesterol and leucine (C + L) (*p* < 0.001) [14]. We also showed that valine and leucine decreased the serum levels of LDL-cholesterol proving lipid-lowering properties [15]. In our experiment we also evaluated the glucose value and we found an important increase when cholesterol is added to the diet, but when the amino acids (valine and leucine) are added, the glycemic values decrease compared to group II.

According to our results, the two amino acids proved their antioxidant abilities, which could improve the endothelial damage related to atherosclerosis [16].

The literature mentions that the appearance of atheroma plaque in the vessel’s intima in rats subjected to a hypercholesterolemic diet occurs in approximately 8 months. In our study, we did not obtain atherosclerotic plaques because the length of the experiment was 60 days, which is not long enough to produce severe lesions. Instead, at the biochemical level, we found changes in the parameters of lipid metabolism and oxidative stress that characterize atherosclerosis in the pre-lesional phases. In addition, according to our results, valine and leucine added to this diet have a direct influence on the lipid metabolism parameters by lowering the level of triglycerides, total cholesterol and LDL-cholesterol. Comparing the two essential amino acids, we noted that valine acts more promptly and rapidly than leucine. Therefore, we assume a possible hypolipemiant action and a consequently anti-atherogenic action of the two compounds.

## 4. Discussion

Triglycerides are present in the lipoprotein particles known as chylomicrons, which contain the largest amount of lipids absorbed in the intestines as well as in the very-low-density lipoproteins (VLDL) containing triglycerides synthesized in the liver. An increased amount of any of these two leads to elevated levels of serum triglycerides and to hypertriglyceridemia as a clinical condition, which is a heterogeneous ensemble of features, each of them contributing to a certain degree in the increase of the total cardiovascular risk [20].

In the present experimental model, we aimed to study the antiatherogenic potential of 2 amino acids (valine and leucine), based on the physiopathological mechanisms of atherosclerosis, in conditions of induced hypercholesterolemia via a cholesterol-rich diet. We analyzed many animal models discussed in the literature [21] and we chose the atherosclerosis experimental model proposed by Anitschkow in 1913 [22].

Amino acids are essential precursors for the synthesis of numerous molecules playing a major role in homeostasis [23,24]. Leucine, valine and isoleucine are branched-chain essential amino acids involved in protein biosynthesis as well as in regulating the cell-division cycle. Leucine, in particular, participates in growth and development of cells in a mTOR-dependent manner [25].

In a study performed in 2018, we assessed the role of valine, leucine and isoleucine on the occurrence and progression of atherosclerosis in rats receiving hypercholesterolic diet. The comparative study of the three essential amino acids revealed that valine induced a faster response than leucine and isoleucine on the improvement of biochemical parameters, but no significant differences between the three amino acids in terms of their protective ability, according to the histopathological lesion assessment [26]. Still, further studies in order to assess the precise molecular mechanism by which these amino acids influence the triglyceride levels are necessary.

In the last years, numerous theories have been issued with regard to the role of essential amino acids in the modulation of atherosclerotic pathophysiology. However, the exact impact and timing of these amino acids intervention are not fully elucidated, as research has so far yielded contradictory results [27].

At present, researchers are engaged in a discussion about both, a possible proatherogenic effect and a potential antiatherogenic role of branched-chain amino acids [28].

Unlike our results, Bhattacharya et al. presented the idea in 2013 that branched-chain amino acids are responsible for increasing cardiovascular-related mortality, being associated with extreme forms of ischemic heart disease. They were able to demonstrate that the relationship is there even after the more traditionally accepted risk factors, such as diabetes mellitus or insulin resistance have been corrected [29].

In a study from 2016, Ruiz-Canela hypothesized that elevated serum levels of branched-chain amino acids as valine, leucine and isoleucine correlate with increased global cardiovascular risk which may not be altered by dieting [30]. In addition, Sun et al. demonstrated via an experimental study on mice that defective catabolism of branched-chain amino acids mediated by Kruppel-like factor 15 (KLF 15) is responsible for cardiac depression manifestations [31].

Based on the results of a prospective cohort study over a period of 18.6 years, Tobias et al., were able to conclude that the connection between branched-chain amino acids and cardiovascular disease is similar to the causal relationship between plasma levels of LDL cholesterol and cardiac mortality [32].

On the other hand, numerous published studies support the beneficial effects of branched-chain amino acids in regulating lipid metabolism and functional cardiac parameters.

Similar to our results, Noguchi et al. in 2006, highlighted the role that valine and leucine play in counteracting the impact of abnormal lipid concentration, directly involved in the production of atherosclerotic lesions [33].

Terakura et al. demonstrated that supplementing the diet with branched-chain amino acids decreases the hepatic triglycerides accumulation and the chronic inflammatory process associated with obesity, most likely by inhibiting interleukin-6 (IL-6), Tumour Necrosis Factor-alpha (TNF-alpha) and monocyte chemoattractant protein-1 (MCP-1) expression. Also, in mice fed with branched-chain amino acids, their average adiposity was lower, probably mediated by peroxisome proliferator-activated receptor gamma (PPAR-gamma) [34].

In 2012, Chen et al. shared their results regarding the influence of leucine on body weight and blood lipids. They witnessed a regulation of carbohydrate and lipid metabolism regardless of how the amino acid was administered (orally or intracerebroventricularly) [35].

Pedroso et al. showed that in rats with metabolic syndrome, restricting their caloric intake concurrent with adding leucine to their diet facilitates an improved protein anabolism, as well as an increase in the levels of leptin and IL-6. The study drew attention to the fact that supplementing the diet with branched-chain amino acids modifies the hepatic metabolism by influencing the metabolism of fatty acids and cholesterol [36].

Another study confirming the role of leucine in lowering the levels of triglycerides and LDL and in raising HDL-cholesterol in diabetic rats to which food was supplemented with this amino acid was published by Sadri in 2017 [37]. However, the therapeutic effect of norleucine is notably inferior to that of leucine. A possible increase in the level of leptin as a result of a leucine-rich diet has been discussed in the literature. The leptin levels obtained, however, have not proven to be statistically significant [38].

Therefore, research results are contradictory and different views are being presented. Some point to the effects of branched-chain amino acids with regard to an increasing in the global cardiovascular risk, while numerous studies highlight the benefits of supplementing the diet with leucine, valine and isoleucine in order to lower the lipid metabolism parameters. Our study evidently supports the latter by showing that leucine and valine lower the level of plasma triglycerides, thus having a positive effect on lipid balance and, consequently, on the integrity of the vascular wall.

## 5. Conclusions

By comparing the measured triglyceride values across different groups according to the experimental design, the biochemical analysis revealed the fact that essential amino acids such as valine and leucine lower the level of triglyceride. Consequently, the vascular endothelium is protected and the risk of endothelial dysfunction diminished.

The comparison of the two essential amino acids indicated that valine acts more promptly and rapidly than leucine. The results of this experiment support the idea that valine and leucine play a distinct and specific role in the evolution of induced atherosclerosis. How these two amino acids behave affords several subsequent research avenues in terms of therapeutic goals, and our study is an attempt to highlight a potential novel therapeutic strategy.

## Figures and Tables

**Figure 1 medicina-57-00239-f001:**
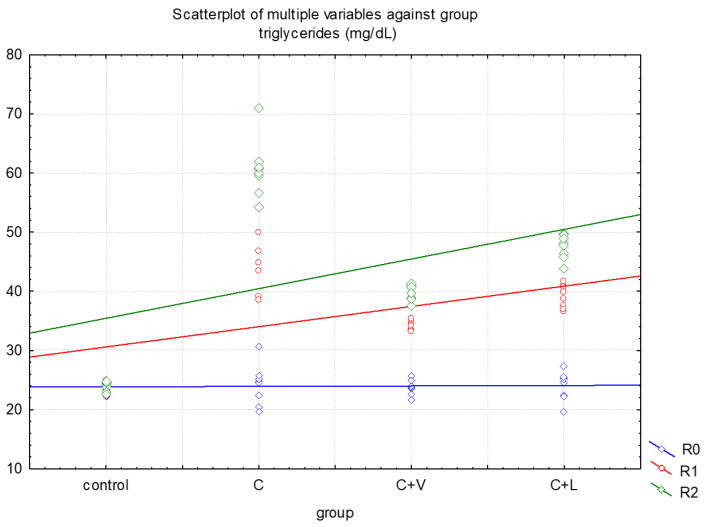
Individual levels of triglycerides in all groups.

**Figure 2 medicina-57-00239-f002:**
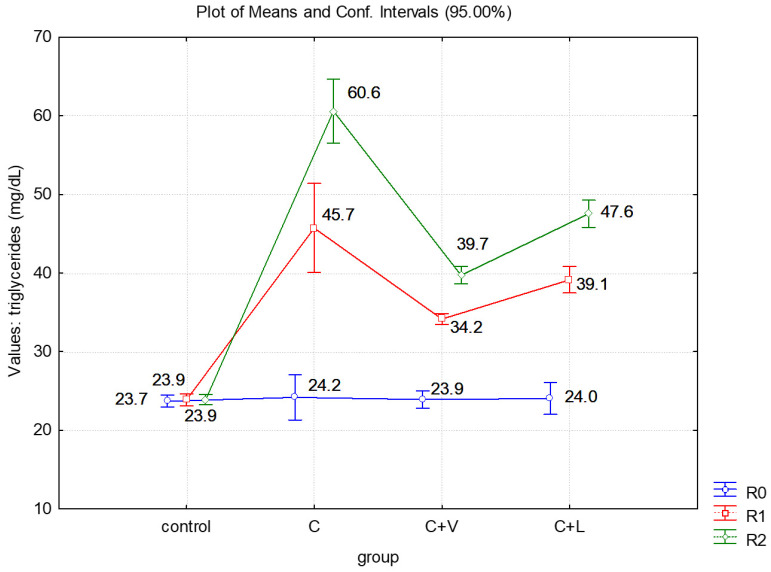
Average levels of triglycerides in all groups.

**Table 1 medicina-57-00239-t001:** Levels of triglycerides in all groups.

Group	R0	R1	R2
**Group I: control**			
**Average mg/dL**	23.70	23.88	23.90
**SD**	0.89	0.90	0.78
**CV%**	3.76	3.77	3.27
**Group II: C**			
**Average mg/dL**	24.19	45.73	60.61
**SD**	3.44	6.78	4.87
**CV%**	14.22	14.82	8.04
**Group III: C + V**			
**Average mg/dL**	23.91	34.15	39.73
**SD**	1.31	0.84	1.30
**CV%**	5.48	2.45	3.27
**Group IV: C + L**			
**Average mg/dL**	24.04	39.15	47.56
**SD**	2.43	2.01	2.07
**CV%**	10.13	5.14	4.36

**Table 2 medicina-57-00239-t002:** Multivariate tests using one-way repeated measure analysis of variance.

Multivariate Tests ^d^									
Effect		Value	F	Hypothesis df	Error df	Sig.	Partial Eta Squared	Noncent. Parameter	Observed Power ^b^
Time	Pillai’s Trace	0.985	879.337 ^a^	2.000	27.000	0.000	0.985	1758.674	1.000
	Wilks’ Lambda	0.015	879.337 ^a^	2.000	27.000	0.000	0.985	1758.674	1.000
	Hotelling’s Trace	65.136	879.337 ^a^	2.000	27.000	0.000	0.985	1758.674	1.000
	Roy’s Largest Root	65.136	879.337 ^a^	2.000	27.000	0.000	0.985	1758.674	1.000
Time * Type	Pillai’s Trace	1.058	10.487	6.000	56.000	0.000	0.529	62.920	1.000
	Wilks’ Lambda	0.028	44.831 ^a^	6.000	54.000	0.000	0.833	268.989	1.000
	Hotelling’s Trace	31.694	137.340	6.000	52.000	0.000	0.941	824.037	1.000
	Roy’s Largest Root	31.596	294.898 ^c^	3.000	28.000	0.000	0.969	884.694	1.000

^a^ Exact statistic. ^b^ Computed using alpha = 0.05. ^c^ The statistic is an upper bound on F that yields a lower bound on the significance level. ^d^ Design: Intercept + Type. Within Subjects Design: Time. * Statistical difference between time and type (groups).

**Table 3 medicina-57-00239-t003:** Pairwise comparisons between the studied groups.

Pairwise Comparisons								
(I) Time		(J) Time		Mean Difference (I-J)	Std. Error	Sig.^a^	95% Confidence Interval for Difference ^a^	
							Lower Bound	Upper Bound
dimension1	R0	dimension2	R1	−11.766 *	0.459	0.000	−12.934	−10.597
			R2	−18.986 *	0.454	0.000	−20.143	−17.829
	R1	dimension2	R0	11.766 *	0.459	0.000	10.597	12.934
			R2	−7.220 *	0.321	0.000	−8.038	−6.401
	R2	dimension2	R0	18.986 *	0.454	0.000	17.829	20.143
			R1	7.220 *	0.321	0.000	6.401	8.038

Based on estimated marginal means. *. The mean difference is significant at the 0.05 level. ^a^ Adjustment for multiple comparisons: Bonferroni.

**Table 4 medicina-57-00239-t004:** Statistical differences between the average levels of triglycerides in all groups.

Time	Group	Control Group (*n* = 8)	C (*n* = 8)	C + V (*n* = 8)
R0	C (*n* = 8)	*p* = 0.737	-	
	C + V (*n* = 8)	*p* = 0.884	*p* = 0.849	-
	C + L (*n* = 8)	*p* = 0.814	*p* = 0.919	*p* = 0.928
R1	C (*n* = 8)	*p* < 0.001 *	-	
	C + V (*n* = 8)	*p* < 0.001 *	*p* < 0.001 *	-
	C + L (*n* = 8)	*p* < 0.002 *	*p* = 0.001 *	*p* = 0.0422 *
R2	C (*n* = 8)	*p* < 0.001 *	-	
	C + V (*n* = 8)	*p* < 0.001 *	*p* < 0.001 *	-
	C + L (*n* = 8)	*p* < 0.001 *	*p* < 0.001 *	*p* < 0.001 *

Post-hoc analysis: Newman–Keuls test; (*) Marked differences are significant at *p* < 0.05.

**Table 5 medicina-57-00239-t005:** Mean values of HDL, LDL and glucose at R0, R1 and R2.

	Group	R0	R1	R2	*p*-Value
**Cholesterol I** **mean ± SD**	Control group	37.14 ± 2.56	37.50 ± 2.09	37.61 ± 1.45	0.0783
	C (*n* = 8)	36.41 ± 4.15	49.89 ± 3.99	76.61 ± 3.46	<0.001 *
	C + V (*n* = 8)	36.67 ± 1.28	41.12 ± 1.27	44.87 ± 1.22	0.001 *
	C + L (*n* = 8)	36.50 ± 2.70	46.04 ± 2.71	49.53 ± 2.12	<0.001 *
	***p*** **-value**	**0.577**	**0.006 ***	**<0.001 ***	
**HDL** **mean ± SD**	Control group	23 ± 1.48	22.88 ± 1.22	22.89 ± 1.68	0.911
	C (*n* = 8)	22.43 ± 3.29	19.44 ± 1.45	15.93 ± 1.20	0.004
	C + V (*n* = 8)	22.98 ± 1.48	24.64 ± 2.79	26.85 ± 2.95	0.114
	C + L (*n* = 8)	22.51 ± 2.15	22.97 ± 1.90	23.17 ± 1.81	0.523
	***p*** **-value**	**0.637**	**0.001 ***	**<0.001 ***	
**LDL** **mean ± SD**	Control group	9.39 ± 3.39	9.83 ± 2.52	9.93 ± 2.71	0.749
	C (*n* = 8)	7.73 ± 4.54	21.3 ± 3.64	47.94 ± 5.47	<0.001 *
	C + V (*n* = 8)	8.9 ± 2.01	9.64 ± 2.79	10.07 ± 2.75	0.486
	C + L (*n* = 8)	9.17 ± 3.43	15.23 ± 2.73	16.84 ± 2.28	0.0004 *
	***p*-value**	**0.319**	**0.001 ***	**<0.001 ***	
**Glucose** **mean ± SD**	Control group	122.75 ± 5.87	122.61 ± 5.84	122.77 ± 6.08	0.962
	C (*n* = 8)	122.78 ± 8.77	149.26 ± 7.73	162.82 ± 5.83	<0.001 *
	C + V (*n* = 8)	121.98 ± 4.71	140.93 ± 4.84	142.37 ± 4.70	<0.001 *
	C + L (*n* = 8)	121.93 ± 5.13	143.79 ± 6.32	148.08 ± 4.78	<0.001 *
	***p*-value**	**0.994**	**<0.001 ***	**<0.001 ***	

Post–hoc analysis: Newman–Keuls test; (*) Marked differences are significant at *p* < 0.05. SD–standard deviation.

## Data Availability

The data presented in this study are available on request from the corresponding author.

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
