# Peer review of "Hypolipemiant Actions and Possible Cardioprotective Effects of Valine and Leucine: An Experimental Study"

_medicina, 2021, doi:10.3390/medicina57030239_

Round 1
Reviewer 1 Report
Summary. In the manuscript entitled "Hypolipemiant actions and possible cardioprotective effects of valine and leucine - an experimental study", Elena Cojocaru et al. aimed to demonstrate the antiatherogenic potential of 2 essential amino acids (valine and leucine) in a rat model with cholesterol-rich diet-induced hypercholesterolemia.
The authors demonstrate that valine and leucine have a direct impact on the lipid metabolism parameters by lowering the level of triglycerides.
Overall, this is a good report presenting several sets of novel and highly relevant results, which meaningfully improves the knowledge of branched-chain amino acid impact upon cardiovascular pathology. However, in its current form, this study has several minor issues that should be carefully addressed.
Critiques:
As a general rule, all abbreviations/acronyms should be written out in full on first use. The authors do not follow this rule in several places. For example, terms like LDL, apoB, HDL, CV, VLDL, TNF, MCP-1, PPAR should be present in accordance with the rule.
The main problem of the research is the statistical analysis that has been performed. The authors applied Student's t-test (by the way, Student's but not Student as they mentioned), which is not appropriate in this case. The t-test can be used to determine if the means of two sets of data are significantly different from each other. Of note, the authors examined four groups of animals; therefore, analysis of variance (ANOVA) test had to be applied here. Moreover, considering that the authors measured the levels of triglycerides throughout the experiment at three different time points, the repeated measures ANOVA should be applied. Of note, repeated measures analysis of variance is a statistical approach to repeated measure designs (just like in the present study). With such designs, the repeated-measure factor is the within-subjects factor, while the quantitative dependent variable on which each participant is measured is the dependent variable.
In addition, figure 1 should be present as a bar plot with individual values and figure 2 as a graph reflecting repeated measures significance. Moreover, the analyses present in figure 2 are not appropriate due to Student's test applied. I believe that the plot is not informative confusing, and unnecessary. The authors can mention ratios in the text. Anyway, repeated measures ANOVA should be applied here for the analysis.
Figure 3 - the sign of significance is absent.
Author Response
Dear reviewer,
Thank you for all your comments.
We modified and introduced all the abbreviations. Thank you for the mentioning.
We reevaluated the statistics and applied ANOVA test (repeated measures ANOVA). We will keep in mind the difference between the two tests and remember it for the future.
We changed Figure 1 and modified it in a bar plot. Instead of figure 2, we added two more tables with the repeated measures ANOVA test. In figure 3, we added the statistical significance sign.
Hope we have touched all the points you asked us to change.
If there are any other changes you consider we should make, please let us know.
Yours sincerely,
All the authors
Reviewer 2 Report
This manuscript presents serum triglycerides data on rats fed with cholesterol containing diets or cholesterol diets supplemented with valine or leucine. The authors showed that while cholesterol diets lead to elevated triglycerides, valine or leucine supplement appeared to partially reduce triglycerides elevation induced by cholesterol diet. Unfortunately, the only data presented here are serum triglycerides levels with no any evidence regarding cardiprotection. Also I would not call the data here indicate lipid lowering effects as triglycerides levels still elevate with valine or leucine supplement compared to just control diet. The authors should include data from valine or leucine alone groups in order to make such a claim. Altogether, this study is incomplete and needs further investigation.
Major:
- Please include at least two more groups in your study (i.e. valine only, leucine only)
- Please show body weight data and more blood lipid parameters (e.g. total cholesterol, VLDL, LDL, HDL) and related data (e.g. glucose levels, cytokine levels, RBC count, WBC count, etc) in order to get a better picture of how leucine or valine supplement affect lipid metabolisms in rats.
- What about other essential amino acids supplement? How do you know the effects you showed here are specific to valine or leucine?
- No evidence here indicates the potential molecular mechanism of how the two amino acids in food could possibly affect triglycerides levels.
Author Response
Dear reviewer,
Thank you for all your comments.
- In the study protocol has been used a single control group, that did not receive any amino acids or cholesterol. Because the design of the study consisted in quantifying the effect of amino acids on cholesterol and its derivatives, it was not considered useful at that time to add a group of amino acids only, as the value of cholesterol was normal in those batches. We will consider improving the study in further research, by adding the groups recommended by you.
- We have added a table with HDL-C, LDL-C and glucose in order to reveal the action of amino acids on cholesterol derivatives. These data have already been published in other articles, mentioned in the bibliography.
- According to the producer's recommendations, we have chosen two of the most studied amino acids, to see their effect on the dyslipidemic syndrome. In our future research, we will expand the study by using other essential amino acids.
- Up until now, there is no clear mechanism to explain the effect of amino acids on cholesterol synthesis, opinions being contradictory, fact mentioned in the article.
Hope we have touched all the points you asked us to change.
If there are any other changes you consider we should make, please let us know.
Yours sincerely,
All the authors
Reviewer 3 Report
Cojocaru and colleagues in the present study entitled “Hypolipemiant actions and possible cardioprotective effects of valine and leucine - an experimental study” investigated the anti-atherogenic potential of two amino acids namely leucine and valine using a diet-induced hypercholesterolemia rat model. The authors observed significantly reduced levels of serum triglycerides in rat groups treated with leucine and valine compared with the control group. They concluded that valine and leucine have a direct effect on lipid metabolism in rats. The study is well-designed and straightforward. However, I have the following minor comments:
- The manuscript has numerous grammatical errors and needs critical English language editing.
- What were the criteria to select these doses of valine and leucine?
- Did the authors measure total cholesterol levels in rat plasma? What were the weights of these rats after 60 days of treatment? Did the authors investigate blood glucose levels in these rats?
Author Response
Dear reviewer,
Thank you for all your comments.
- The article will be sent to the MDPI English Editing Services for the English language editing.
- The valine and leucine doses have been chosen according to the producer’s recommendations.
- We have added a table with total cholesterol, HDL-C, LDL-C and glucose in order to reveal the action of amino acids on cholesterol derivatives. These data have already been published in other articles, mentioned in the bibliography. The study protocol did not consider weighing the study in experimental animals.
Hope we have touched all the points you asked us to change.
If there are any other changes you consider we should make, please let us know.
Yours sincerely,
All the authors
Round 2
Reviewer 2 Report
I have reviewed your revision and read your response letter. The major point in this study is the two amino acids, valine and leucine, show some effects on triglycerides with no potential mechanism data. It should be noted that whether and how triglycerides affect atherosclerosis is not clear as well. In addition, I do not see any atherosclerosis data (for example, sizes of the plaques, immune cell infiltration at the aortic sinus, etc) or cardiac functional data. Then whether your study is even relevant to the atherosclerosis or cadioprotective effects are in question.
Author Response
Thank you very much for your suggestions.
The current guideline published in 2020 regarding the atherosclerotic disease describes the involvement of triglycerides and implicitly their value in the pathogenesis of atherosclerosis. Starting from these facts, we tried to identify new compounds useful in reducing triglyceride levels. Why did we choose valine and leucine? Because previous studies reported that both valine and leucine lead to the formation of odd‐chain fatty acids, due to the induction of oxidation dependent on peroxisome proliferator activated receptor alpha and the increased de novo synthesis of fatty acids from propionyl‐coenzyme A. Since the activation of peroxisome proliferator activated receptor alpha accelerates the oxidation of fatty acids, we assume that valine and leucine decrease the concentration of triglycerides by affecting the oxidation of fatty acids.
The literature mentions that the appearance of atheroma plaque in the vessel's intima in rats that undergo a hypercholesterolemic diet occurs in approximately 8 months. In our study we did not obtain atherosclerotic plaques because the length of the experiment was 60 days and therefore not long enough to produce severe lesions. Instead, at the biochemical level, our results showed changes in the parameters of lipid metabolism and oxidative stress that characterize atherosclerosis in the prelesional phases.
We consider that the limitations of the study were represented by the infrastructure that did not allow the experiment to be carried out for more than 60 days, as well as the deficit in modern equipment that would have allowed in-depth analysis of the atherosclerotic process (ultrasound, echo doppler, intima-media thickness of carotid artery, arteriography to determine arterial stiffness).
Hope we have touched all the points you asked us to change.
If there are any other changes you consider we should make, please let us know.
Yours sincerely,
All the authors
Round 3
Reviewer 2 Report
I have no more comments.